# Aggressive Impacts Affecting the Biodegradable Ultrathin Fibers Based on Poly(3-Hydroxybutyrate), Polylactide and Their Blends: Water Sorption, Hydrolysis and Ozonolysis

**DOI:** 10.3390/polym13060941

**Published:** 2021-03-18

**Authors:** Anatoly A. Olkhov, Polina M. Tyubaeva, Alexandre A. Vetcher, Svetlana G. Karpova, Alexander S. Kurnosov, Svetlana Z. Rogovina, Alexey L. Iordanskii, Alexander A. Berlin

**Affiliations:** 1Department of Chemistry and Physics, Plekhanov Russian University of Economics, Stremyanny Ln 36, 117997 Moscow, Russia; aolkhov72@yandex.ru (A.A.O.); polina-tyubaeva@yandex.ru (P.M.T.); 2N.M. Emanuel Institute of Biochemical Physics, Russian Academy of Sciences, Kosygin St. 4, 119991 Moscow, Russia; karpova@sky.chph.ras.ru (S.G.K.); sannygraffitiking@yandex.ru (A.S.K.); 3N.N. Semenov Federal Research Center for Chemical Physics, Russian Academy of Sciences, Kosygin St. 4, 119334 Moscow, Russia; s.rogovina@mail.ru (S.Z.R.); aljordan08@gmail.com (A.L.I.); berlin.a@yandex.ru (A.A.B.); 4Nanotechnology Scientific and Educational Center, Institute of Biochemical Technology and Nanotechnology, Peoples’ Friendship University of Russia (RUDN), Miklukho-Maklaya St. 6, 117198 Moscow, Russia

**Keywords:** poly(3-hydroxybutyrate), polylactide, blends, ultrathin fibers, electrospinning

## Abstract

Ultrathin electrospun fibers of pristine biopolyesters, poly(3-hydroxybutyrate) (PHB) and polylactic acid (PLA), as well as their blends, have been obtained and then explored after exposure to hydrolytic (phosphate buffer) and oxidative (ozone) media. All the fibers were obtained from a co-solvent, chloroform, by solution-mode electrospinning. The structure, morphology, and segmental dynamic behavior of the fibers have been determined by optical microscopy, SEM, ESR, and others. The isotherms of water absorption have been obtained and the deviation from linearity (the Henry low) was analyzed by the simplified model. For PHB-PLA fibers, the loss weight increments as the reaction on hydrolysis are symbate to water absorption capacity. It was shown that the ozonolysis of blend fibrils has a two-stage character which is typical for O_3_ consumption, namely, the pendant group’s oxidation and the autodegradation of polymer molecules with chain rupturing. The first stage of ozonolysis has a quasi-zero-order reaction. A subsequent second reaction stage comprising the back-bone destruction has a reaction order that differs from the zero order. The fibrous blend PLA/PHB ratio affects the rate of hydrolysis and ozonolysis so that the fibers with prevalent content of PLA display poor resistance to degradation in aqueous and gaseous media.

## 1. Introduction

Eco-friendly, fully biodegradable materials—specifically, films with micrometric diapason of thickness and the ultrathin fibers with submicrometric diameters—are in peak demand for a constructive and special performance in such areas as biomedicine [1,2], packaging [3,4], environment protection (eco-friendly absorbents and particularly) [5], etc. In the above-specified areas of application, the engineering transition from micro-structured materials to nanoscale polymer systems has been driven by the emergence of innovative features, such as high surface/volume ratios, enhancements in mechanical behavior, increased impacts of diffusivity impact on drug release and pollution sorption, and the improvement of filtration efficacy for liquid and gaseous media. As electrospun products, scaffolds and nanofibrous composite implants are increasingly being used in tissue engineering and regenerative medicine [6,7]. These medical devices promote and control the growth of cells, tissues, and organs as well as provide the targeted controlled delivery of the wide spectrum of bioactive compounds [8,9]. Along with environmentally friendly, biodegradable barrier films [10,11], biopolymer ultrathin fibers are beginning to be used for the transition from passive conventional packaging to active smart packaging, which is devoted to the enhancement of food’s shelf-life [12,13]. Another area of electrospun fiber application—no less important than the previous ones—is the efficient elimination of disastrous anthropogenic consequences resulting from both local and global spill accidents [14]. For the achievement of the desired water quality, its high-effective separation from oil pollutions can be carried out by nanostructured polymer adsorbents characterized by highly developed surfaces with relevant hydrophilic–hydrophobic ratios [15,16].

In this study, the electrospun ultrathin fibers were based on thermoelastic biopolyesters produced from the naturally-abundant renewable resources, namely, polylactide (PLA) and poly(3-hydroxybutyrate) (PHB) and their blends [17,18,19]. The obtained fibers match the above-mentioned criteria and could be used as nano-structured biodegradable food packaging modifiers and absorbents. During electrospinning, the ultrathin fibers form the film-like fibrous membranes (the mats) with high porosities and large specific surface areas. The high water permeability and high absorption selectivity of these fibrillar membranes regarding the low polar organic substances provide opportunities for their application in water–oil separation systems. An additional important advantage of the proposed PLA-PHB systems, in comparison with traditional polymer absorbents based on polyolefins, polyfluorides, polyamides, etc., is the ability to biodegrade after the end of their service lives, which makes their utilization environmentally safe [20]. Both the single filaments and the fibrillar mats during the exploitation under aqueous environmental conditions are affected by various factors, such as water absorption, temperature evolution, oxygen and ozone impacts, UV radiation, and certainly exposure to microorganisms. These factors can act simultaneously or sequentially depending on the exploitation of aggressive conditions and climatic features.

At the microscopic level, in films and membranes, hydrolysis and enzymatic degradation of biopolyesters such as PHB and PLA as well as their blends have been comprehensively outlined in many recent publications [21,22,23,24,25,26,27,28,29,30,31,32]. Diffusion phenomena and catalytic reactions in the samples at the micro/macro scale were experimentally collected and modeled via kinetic simulations to elucidate the degradation mechanisms and evaluate the exploitation lifespans of the biodegradable materials [24,25,26,27]. A more detailed analysis of the literature will be given in the preamble of Section 3.2, dedicated to the PHB/PLA degradation. However, the analogous challenges of hydrolytic decomposition for the PHB/PLA ultrathin fibers still require an all-encompassing experimental basis as well as mathematical efforts for data processing.

Ozone is an extremely aggressive oxidizer that has a noticeable effect on the chemical structures and molecular dynamics of polymers. Moreover, ozone displays a photooxidative effect at ambient environmental conditions, which was briefly described by some authors recently [33]. Despite the permanent contact of the gas-oxidizer with biomedical and packaging polymers during service, the influences of ozone on the morphology and dynamic characteristics of biodegradable polymers remain a poorly studied topic of polymer science. Despite this, quite recently, the commercialization of PHB and PLA owing to their cost reduction and increasing biomedical demands gave rise to several comprehensive papers devoted to the kinetics and mechanism of the ozonolysis of the biopolyesters [34]. The additional relevant information concerning the two-step consecutive mechanism of ozonolysis is provided in the Section 3.3 of the present publication.

As the initial step of the complex approach to the aggressive impact of external factors, it is necessary to explore the influences of water absorption, hydrolysis, and ozone impacts—that is the aim of this article. This study is focused on the principal aggressive agents modifying the structures and functional behaviors of such biopolyesters as PHB, PLA, and their blends.

## 2. Materials and Methods

In the current research, PHB (BIOMER (Schwalbach am Taunus, Germany)) and NatureWorks^®^ Ingeo™ 3801X Injection Grade PLA (Shanghai Songhan Plastics Technology Co., Ltd. (Shanghai, China)) were studied. PHB was in the form of a powder, obtained by microbiological synthesis with a molecular weight of 460 kDa, a melting point (T_melt_) of 175 °C, and a density of 1.25 g/cm^3^. PLA was in the form of granules with a medium viscosity molecular weight of 1.9 × 10^5^ g/mole. All electropinning solutions were prepared in chloroform (grade CP, PJSC Khimprom, (Novocheboksarsk, Russia)). The optimization of electrical conductivity was achieved by adding into all electropinning solutions the formic acid and the tetrabutylammonium iodide (TBAI) (grade CP, Biochem Chemopharma, (Cosne-Cours-sur-Loire, France)). The mats were manufactured by electrospinning method (ES). In the work, nonwoven fibrous materials obtained by the ES method on a single-capillary laboratory unit with a capillary diameter 0.1 mm were used. The choice of the conditions varied depending on the properties of the chloroform solutions: conductivity, viscosity, and homogeneity. The voltage was 17–22 kV; the distance between the electrodes was 150–200 mm; the gas pressure on the solution was 10–30 kg(f)/cm^2^.

Moisture absorption of mixed fibers was measured by the weight on McBen balances with quartz helix. The sensibility of a quartz helix was 1.5 mm/mg. All samples of the non-woven material were approximately 50 mm × 30 mm and weighed between 60 and 100 mg. The thickness of the sample was measured with a circular indicator with a measurement error of around 1μm using cover glasses for microscopy. The thickness was measured at different points of the sample area; the dispersion of thickness value was 5–10%. Sorption measurements were done on a quartz helix in a thermostatic vacuum column. The column was dried for an hour using a backing vacuum pump at a pressure of 13 Pa. The water vapor was fed to the column with a set pressure after vacuuming. The elongation of the quartz helix due to an increase in the weight of the suspended sample was recorded using a cathetometer. The interval sorption was assessed by repeated vacuuming and filling with steam until the water vapor activity in the vessel reached a value of 0.8–0.9. The water diffusion coefficient was calculated from kinetic curves in each pressure range for the material samples. The sorption isotherm was created based on the interval value of moisture absorption, as the dependence of moisture absorption on the activity of water vapor.

Molecular mobility was studied by a spin probe on the automated EPR-b spectrometer (Institute of Chemical Physics, Russian Academy of Sciences, Moscow, Russia). A stable nitroxide radical 2,2,6,6-tetramethylpiperidine-1-oxyl (TEMPO) was used as a spin probe. The radical was introduced into the sample from the vapor at a temperature of 60 °C, and the concentration did not exceed 10^−3^ M. Registration of EPR spectra was performed in the absence of saturation, which was checked by the dependence of the signal intensity on the power of the microwave field. The values of the probe rotation correlation time τ were calculated from the EPR spectra according to:τ = ∆H_+_ × [(I_+_/I_−_)^0.5^ ‒ 1] 6.65 × 10^−10^(1)
where ∆H_+_—width of the spectrum component in a low field; I_+_/I_−_—the ratio of intensities in low and strong fields. The measurement error (τ) was ±5%.

The study of the thermophysical characteristic of samples was carried out by the DSC method on the DSC 204 F1 (“Netzsch” (Selbu, Germany)) in an Ar medium with a rate of temperature change of 10 °C/min. The average error in thermal effects measurement was approximately ±3%. The value of the degree of crystallinity (χ was calculated according to:χ (%) = 100 × *H_m_/H_PHB_*(2)
where χ—degree of crystallinity; *H_nл_*—fusion heat obtained experimentally; *H_PHB_*—fusion heat of the perfect crystalline of PHB, which is 146 J/g.

The average diameter of the fibers was determined by the SEM on the Hitachi TM-1000 (Tokyo, Japan). Direct measurement of the fiber geometric parameters was performed by the program Adobe Photoshop.CS3.Extended (Adobe, Inc., San Jose, CA, USA). The size distribution was based on the measurements of at least 300 fibers, and the average diameter was determined by the measurements of at least 100 fibers.

Ozonation of fibrous materials was carried out using a flow-type ozonizer (Emanuel Institute of Chemical Physics, Russian Academy of Sciences, Moscow, Russia). The principle of operation of the ozonator was to obtain ozone from oxygen through an electric corona discharge. Ozone was fed into a suspended sample reactor. The gas concentration was controlled by varying the electrical voltage. The experiment was carried out at a working ozone concentration of 5.5 × 10^−5^ mol/L and a temperature of 23 °C. The assessment of the amount of absorbed ozone was carried out using an SF-46 “Lomo” spectrophotometer (Russia) by measuring the optical density of the gas medium leaving the reactor at a wavelength of 254 nm. The gas flow rate was 101.8 mL/min. The analysis of materials after ozonolysis was carried out by DSC and EPR methods using the nitroxyl radical TEMPO. The radical was introduced into the sample from the vapor at a temperature of 60 °C, and the concentration did not exceed 10^−3^ M.

The study of hydrolytic degradation of the materials was carried out in a phosphate buffer that simulates the impact of the physiological environment. The materials were incubated in the phosphate buffer with a concentration of 0.025 M (pH = 7.4) at 70 °C for 21 days. The controlled weighing of materials was performed at regular intervals. The materials were extracted from the buffer solution, washed with distilled water, placed in an incubator for 3 h at 70 °C, and then weighed (the measurement error was 0.1 mg). In experiments with a phosphate buffer, the buffer was replaced every 3 days.

## 3. Results

### 3.1. Water Absorption in Fibrous PLA-PHB Blends

The rate of hydrolytic degradation in the fibrous biopolyesters essentially depends on water absorption and diffusion. At the first stage of hydrolytic degradation, the water molecules are adsorbed on the surface of the fibrils packed in the mats. At the second stage, the penetration of water molecules into the single filament volume occurs, and the chemical and physical interactions of water with the polar centers (functional groups) of the polymer chains are initiated. The water absorption capacity depends on the chemical nature (hydrophobicity–hydrophilicity ratio) of the macromolecules, polymer crystallinity, surface special features, and volume morphology, as well as on the fibril porosity, interfibrillar space of the mats, and other structural factors.

The equilibrium moisture absorption values of the non-woven fibrous materials based on PHB, PLA, and their binary blends were briefly reported earlier in [35,36]. Simultaneously, the corresponding microphotographs of the PHB-PLA fibrils assembled into the mats have shown that the materials consist of cylindrical fibers randomly stacked into each other [37]. Despite both biopolymers, as polyesters, having the same functional groups, the repeated units of PHB and PLA slightly differ from each other in chemical structure, and hence, in water absorption capacity. Moreover, the initial crystallinity of PHB (~70%) is far higher than the crystallinity of PLA (>25–30%), which should affect the total absorption in the homopolymers and the blends [37]. The isotherms of moisture absorption for PHB-PLA blends with the different polymer compositions are shown in Figure 1.

In our case, due to a limited number of experimental points, we have roughly approximated the water absorption isotherms by polynomial function to estimate the deviation of the equilibrium absorption curves from linearity, specifically from the Henry low. Even such a simple expression as Equation (3) has a satisfactorily reasonable concordance between simulation and experiments; see the coefficient regression data in Table 1 below.
G_w_ = B_0_ + B_1_a_w_ + B_2_a_w_^2^(3)

Here, G_w_ is the polymer weight gain; a_w_ is the water activity determined as the relative vapor pressure (p/p_0_); and B_0_, B_1_, and B_2_ are the corresponding polynomial coefficients provided that B_0_ = 0.

The amounts of water gained in the pristine biopolymers PHB and PLA, and in their blends, in each case as a function of water activity, are given in Figure 1A. The experimental points were fitted by Equation (1), and the results of fitting are drawn as solid and dashed curves. The inlet in the Figure 1 illustrates the principal results of absorption data evaluation and confirms the validity of the simplified Equation (4) derived from Equation (3).
G_w_/a_w_ = B_1_ + B_2_a_w_(4)
where all the symbols are the same as in Equation (3).

The estimated coefficients B_1_ and B_2_ with the corresponding correlation coefficients of linear fitting are presented in Table 1. The results given in Table 1 show that the coefficient, B_1_, which is the equivalent of the Henry factor, essentially depends on the fibrous content in the interval from 2.9 × 10^−2^ g/g (PHB) to 5.1 × 10^−1^ g/g (PLA), while the second coefficient B_2_ has a relatively narrow range between 0.41 and 0.74. This remarkable difference in B_1_ could be attributed mainly to both the diversity of polarity for PHB or PLA and high degree of PHB crystallinity preventing segmental mobility, and hence water absorption as a dynamic process. All the blended fibrillar compositions have the values of moisture absorption situated between the values for pristine PHB and PLA. In the mixed fibrillar mats, growth in the PLA concentration led to growth in the water content in the fibrous mats. The B_2_ parameter reflects the deviation of water isotherms from linearity, due to the mutual interaction between water molecules mainly. Such an approximate approach does not contradict the main ideas of the two above-mentioned models—BET and the more advanced GAB—which both describe the two populations of absorbed water interacting with macromolecules and other molecules.

For the specimens of the PHB/PLA blends, the obvious diversities in water sorption (Figure 1) and ozonolysis rate (Figure 6) resulted from the definite difference in polarity of the two repeated polyester units, namely, isobutyric acid (IBA) and lactic acid (LA), respectively. For PLA, owing to the higher proportion of hydrophilic entities and the major impact of terminal groups upon the total polarity of the polymer molecule, water’s affinity to the LA monomer is higher compared to its affinity to IBA. Moreover, the crystallinity of PHB (~70%) is far higher than the crystallinity of PLA (>25–30%), which should affect the total absorption in the homopolymers and the blends dramatically [37]. Therefore, in the blends, the shift of polymer ratio from PHB to PLA led to the water content increasing incrementally.

### 3.2. Weight Loss of PHB/PLA Fibrous Mats

Hydrolytic degradation of PHB and PLA films and membranes has been considered in comprehensive detail in the works of Bonartsev et al. [21,22]; Arrieta, Kenny et al. [23,24,25]; Cameron et al. [26]; and many other prominent authors; see, e.g., [27,28,29,30,31,32]. Note that the acid hydrolysis mechanism for the ester functional groups is significantly different from the process in the alkaline medium [38,39]. Modeling of the combination of hydrolytic and diffusional processes for homogeneous and heterogeneous media was presented recently in the works [40,41,42]. However, in the literature, less attention has been paid to the hydrolytic decomposition of ultrafine fibers and nanofilaments [40,41,42,43]. Finally, hydrolysis of nano-structured fibrillar PHB/PLA composites has been considered in an extremely limited number of works [21,37,44,45,46]. Please note that all the above references are only for hydrolytic degradation cases and do not include enzymatic or cell biodegradation, where more progress has been made [34,47,48]. The phenomenological consideration of the consequences of aqueous hydrolysis of PHB/PLA fibrillar blends, and pristine biopolymer fibers and mats are presented in this section.

It is well known in the literature that after electrospinning, the crystalline and amorphous structures of polymer fibers often are non-stable. That is especially typical for biodegradable polymers whose structures are most vulnerable to physical and chemical effects [49]. The non-equilibrium structure of the biopolymer fibers is named physical aging, and the specific behavior is especially apparent in terms of water absorption and diffusivity. Both fundamental processes determine the chemical resistance of the fibrous materials when the PHB and PLA mats, and their blends, are exposed to aqueous aggressive media. The kinetic dependence for the weight loss of PHB-PLA fibrous mats on the time exposure in the phosphate buffer as the hydrolysis criterion is shown in Figure 2.

Figure 2 depicts that the most intensive hydrolytic degradation was observed for the mats with high concentrations of PLA—over 50%. In 21 days of experimentation, the fibrous blends containing less than 50% of PLA lost their mass, but less than 10%, being relatively stable systems. The samples containing the lower concentrations of PLA (100:0, 10:90, and 50:50 wt.%) demonstrated great weight loss with a short induction period within 3 days. However, the induction periods for the kinetic curves of materials with concentrations of PLA less than 50% are twice as large—about 6 days. During this period, the fibers of pristine PLA lost up to 40% of their initial weight. As was mentioned in the previous section, owing to the lower crystallinity, lower molecular weight, and higher hydrophilicity of repeated units for PLA, its water absorption is almost three times higher than that of PHB fibers. According to DSC data obtained by the authors recently [50], the degree of PLA crystallinity in the fibrillar structures is approximately 2.5 times less than the degree of crystallinity of PHB fibers, which promotes the acceleration of hydrolysis for the PLA fibrous mats compared to the PHB mats.

In Figure 3 the consequences regarding the PHB/PLA mats’ morphologies before and after 21 days are compared. It is clearly seen that after 21 days of exposure to phosphate buffer, a drastic hydrolytic effect demonstrated individual filament disintegration into shorter fragments, which transformed the system, giving it extra brittleness. At the same time, the degradation degree of the fibers with the PLA content predominance (>50%) was more than 50% higher than the samples enriched with PHB [22,51].

In the conclusion of this section, its immediate relationship with the preliminary section should be noted. Objectively, for PHB-PLA fibers, the loss weight incremented because the hydrolysis (see Figure 2) was symbate to water absorption capacity in the same fibrous samples.

### 3.3. Ozone Oxidation

The ozonolysis affects the structures of the polymers in a complex way. At least two consecutive processes had an impact on morphology and molecular dynamics in the biopolymers (Figure 4). Firstly, the network formation via pendant groups occurred at favorable conditions, especially in the intercrystalline polymer space where elongated chains closely contacted each other. The relevant scheme (A) of PLA ozonolysis was recently advanced by Olevnik-Kruszkovska, Novaczyk, and Kadac [34]. As a result of conformational restriction for straightening segments, a decrease in segmental mobility occurs that leads to the reduction in rotation diffusivity of the ESR probe [52]. With the further continuation of ozone exposure, destruction of the main polyester chain, particularly for PHB, accompanied by a decrease in averaged molecular weight, occur [53], resulting in a segmental mobility increment owing to removing the conformational constraints after back-bone disruption.

To assess the kinetic consumption of ozone by the pristine homopolymers PLA and PHB, and their blends formed as electrospun fibers, the kinetic curves were obtained by the preliminary elaborated method in a flow reactor under steady-state conditions [54]. Figure 5 displays the temporal course of ozone absorption for the PLA-PHB fibrous blends with different PLA:PHB wt.% values.

All the curves, except the PHB one, have a similar form—namely, until about the fiftieth minute, the kinetics are satisfactorily described by the linear approximation, and then their form changes noticeably because of the rate of ozonolysis increasing. For the ozone consumption being registered for the PHB, the inflection point is absent and the total kinetic profile of O_3_ absorption is monotonic. The two-stage character of the consumption is very likely to reflect both types of consecutive reactions—the creation of pendant product-ozonation groups (reaction A) and the self-degradation of polymer molecules with their chain ruptures (reaction B). The first stage of ozonolysis has a quasi-zero-order reaction that is consistent with the results of the previous pioneering work [34]. A subsequent second reaction stage comprising the back-bone destruction has a reaction order that differs from the zero-order.

The A and B fragments of Figure 6 present the initial (by ~50 min) and the corresponding kinetic constants presented as functions of the biopolymer ratio. As follows from Figure 6B, the ozonolysis constant dependence has a clear maximum at a 50:50 ratio of constituent components, PHB and PLA. The increase in absorption rate by the PHB-PLA blends in comparison with absorption by the homopolymers is determined by the drop of crystallinity in the system, and most notably in PHB as the result of blending. Hence, higher accessibility of amorphous areas to ozone should occur. As shown before for the fibrillar and plane film composites of PHB-PLA [55], in the range of component ratio 30:70–70:30, their crystallinity was decreased, which reflected an enhancement in segmental dynamics—namely, at those polymer ratios, the time correlation values have a flat minimum (see Figure 4 in [47]). Besides, in this PHB concentration interval, the diffusivity of the low-molecular-weight compound—in particular, the modeling drug, dipyridamole—was increased, which elucidated the mechanism of controlled release in PHB-PLA electrospun fibers [37].

The effect of ozonolysis duration on molecular dynamics in PHB-PLA fibers is demonstrated in Figure 7. Here, the evolution of the correlation time (τ) was evaluated from the ESR spectra characteristics (see formula A below). The values of τ reflect the rotation mobility (diffusivity) of the molecular probe, TEMPO, embedded in the fiber volume [56].

Within the first 50 min of ozonolysis—at the initial stage of the kinetic curves belonging to the composite fibers with high PHB content—the rise of τ was observed, which is consistent with the above-suggested mechanism of network formation via oxidized pendant groups and the ensuing cross-linking. For pristine PHB, due to its high crystallinity and the narrow amorphous space, the values of τ are practically constant, which is due to the low ozone penetration into PHB amorphous area with relatively high density. For the same sample, the next stage, namely, the macromolecular destruction, manifests itself rather weakly, without the essential impact on the general process of oxidation. As the second component, PLA, begins to prevail in the PHB-PLA blends, the τ decreases, and correspondingly, the enhancement in segmental mobility becomes quite noticeable (see curve 3). The intensity of probe rotation in the amorphous space only (the Tempo probe does not penetrate into crystalline compounds due to its steric size) depends on the PHB-PLA ratio in the electrospun fibers. With decreasing PHB content, the kinetic curves in Figure 7 demonstrate a general decrease in ESR probe mobility (see curves 4–6).

The forced orientation of polymer molecules under the action of external factors was established first by Flory [57]. Currently, a lot of polymer systems have been successfully applied in innovative areas of oriented-materials, namely, for enhanced thermoelectric performance [58], stimuli-responsive technology platform design [59], polymer folding-unfolding transition study [60], and 3D and 4D-printing in tissue engineering [61]. Previously, the phenomenon of orientation for macromolecules exposed to ozone was established in the studies [62,63].

During electrospinning under electrodynamic and viscoelastic forces, the blended polymer’s jet in a gel-like status comprises the macromolecules which are partly oriented along the fiber drawing axis. During the jet’s solidification and the following transition to a glassy-state, PHB and PLA molecules incompletely lose the orientation while the rest of the polymer molecules retain forced straightening.

Taking into account the partial segmental orientation for electrospun PHB/PLA fibers, during the initial stage of oxidation, the biopolymer’s rigidity increases because of the formation of polar groups and the following physical-chemical intermolecular cross-linking, e.g., via H-bonds, with the loss of segmental mobility. The perfect segmental packing is more typical for the homopolymers and interrupted in the binary blends due to the poor compatibility of PHB and PLA. Figure 7B confirms this thesis because the activation energy of probe radical rotation depends on the polymer content, and the activation barrier is minimal for the biopolymer blends in the same concentration range where the ozonolysis constants are maximal. Hence, the lower packaging density in the amorphous areas of the blends determines the higher rate of ozone oxidation when compared to the pristine homopolymers PHB and PLA, wherein the cohesion of polymer molecules should be more significant. Simultaneously, we can realize that rate of ozonolysis in the PLA/PHB blends is directly conditioned by the segmental mobility of the biopolymer chains.

As it was shown in Figure 1, the water sorption was changed monotonically according to the PHB/PLA proportion, while the ozonolysis rate dependence (Figure 6B) had the extremum at a 50/50 weight ratio, which could be related to poor compatibility between the biopolyesters, and hence, due to the increase in the surface/volume ratio of the blend. In previous works, we have already observed the extremal behavior of drug diffusivity, segmental dynamics in PHB/PLA, and PHB/polyamide systems [64,65] at such polymer ratios, whereat the separation of the polymer phases are maximal. The ozonolysis efficacy is proportional to the total ozone flux and hence to the extent of interphase through which ozone molecules penetrate the functional groups. In the near future, we are going to determine the specific surfaces of fibrous polymers by Ar absorption technique and compare the ozonolysis activity with surface development as a test of polymer incompatibility.

The contemporary level of the development of polymer theory allows predicting some configurations and dynamics of segments based on the theory of fractals. It is a very important tool that could be used to predict the structures, morphologies, and segmental dynamic behaviors of fibers [66,67]. However, such calculations still require careful examination of the relevance of their predictions to the behavior of real polymer blends. Only after collection and analysis of extensive experimental data, will we be able to rely on them.

## 4. Conclusions

The negative impacts of petrol-based polymers upon the ecosystem can be successfully reduced by the implementation of natural biodegradable plastics that, hopefully, will completely replace the synthetic materials currently in force. Ecosystems especially suffer from waste such as nonbiodegradable food packaging and household packing. Bio-based and biodegradable polyesters, namely, PHB, PLA, and their blends, as biomedical, eco-friendly packaging materials, have been attracting considerable academic and industrial attention. The workability during exploitation and weathering during landfill storage could be accompanied by hydrolysis and/or ozone oxidation. This area of investigation and practical usage has a still relatively poor experimental basis that is especially true for the ultrathin fibrillar biopolyesters.

The direct effect of humidity modulated by the PHB/PLA content on the rate of hydrolysis was interpreted in the framework of segmental dynamics. The latter characteristic was evaluated by the TEMPO-probe ESR technique to record the time correlation (τ) that reflects averaged molecular mobility in the fibers. Objectively, for PHB-PLA fibers, the loss weight increment as the reaction on hydrolysis is symbate to water absorption capacity in the same fibrous samples. The most intensive hydrolytic degradation was observed for the mats with high concentrations of PLA—over 50%. The current study demonstrated the two-stage kinetic mechanism of ozonolysis in ultrathin fibers, which includes the pendant group formation and the following molecular degradation, which includes backbone disruption. The two-stage character of the consumption very likely reflects two types of consecutive reactions: the creation of pendant product-ozonation groups and the self-degradation of polymer molecules with chain rupturing.

Thus, the water sorption occurs in the bulk of the fibers and as the equilibrium process determined by the thermodynamic affinity of water molecules to polymer molecules. Generally, the change in affinity, the effect of the terminal groups, and the polymer crystallinity lead to the monotonic shift of water content when PHB is replaced by PLA. Since hydrolysis is accompanied by water molecules, the water concentrations in the blends influence the rates of hydrolysis that are manifested in the simbate increases of the coefficient B1. In contrast, ozonolysis, tentatively, proceeds with the participation of ozone diffusion, which is enhanced by phase separation of the blends owing to the interphase increase. The maximal phase separation occurs when the polymer content is equal to 50/50, where the rate of ozonolysis is maximal as well. For the ultrathin fibers with PLA predominance, the faster hydrolysis and ozonolysis result from higher polarity and lower crystallinity as compared with PHB. These characteristics must cause the water sorption increment and enhanced reaction capacity specific to ozone.

The presented results support the necessity for the further implementation of biodegradable materials in eco-friendly products—e.g., to create completely biodegradable porous packaging material or eco-friendly fibrillar sorbents. Additionally, based on the results, future research will be focused on UV degradation and biodegradation of the PHB-PLA fibrillar systems.

## Figures and Tables

**Figure 1 polymers-13-00941-f001:**
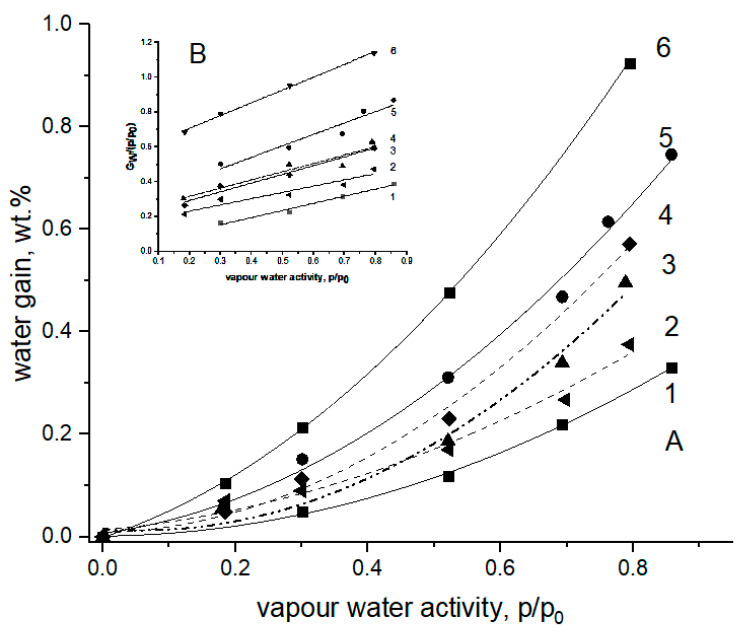
The isotherms of moisture absorption for pristine PHB and PLA fibers and their fibrillar blends at 25 °C [36]. (**A**) The PHB/PLA weight ratios: 1—100/0 (PHB), 2—70/30, 3—50/50, 4—30/70, 5—10/90, 6—0/100 (PLA). (**B**) Linearization of experimental data (points) by the polynomial equation—Equation (1).

**Figure 2 polymers-13-00941-f002:**
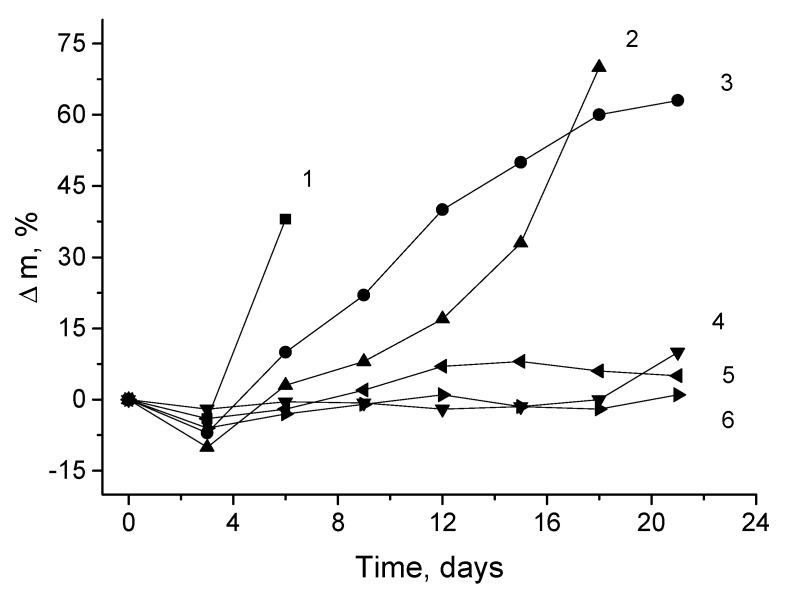
The biopolymer weight loss kinetics during exposure in the phosphate buffer, where PHB:PLA (wt.%): 1—0:100; 2—10:90; 3—50:50; 4—70:30; 5—90:10; 6—100:0.

**Figure 3 polymers-13-00941-f003:**
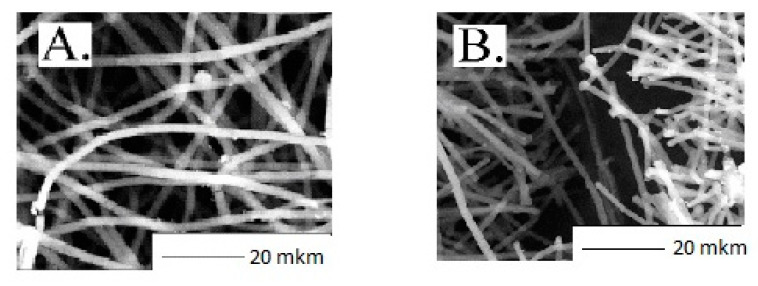
Microphotographs of the non-woven materials based on PHB: (**A**) before and (**B**) after 21 days of hydrolysis in the phosphate buffer solution.

**Figure 4 polymers-13-00941-f004:**
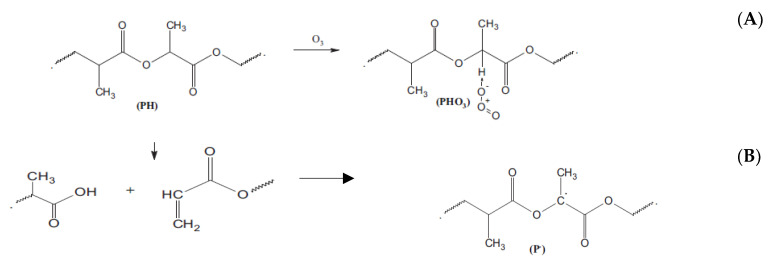
The scheme of polyester (PLA) ozonolysis. Two principal consecutive reactions of pendant groups’ formation (**A**) and the chemical rupture of the polymer chain (**B**)—advanced by [34].

**Figure 5 polymers-13-00941-f005:**
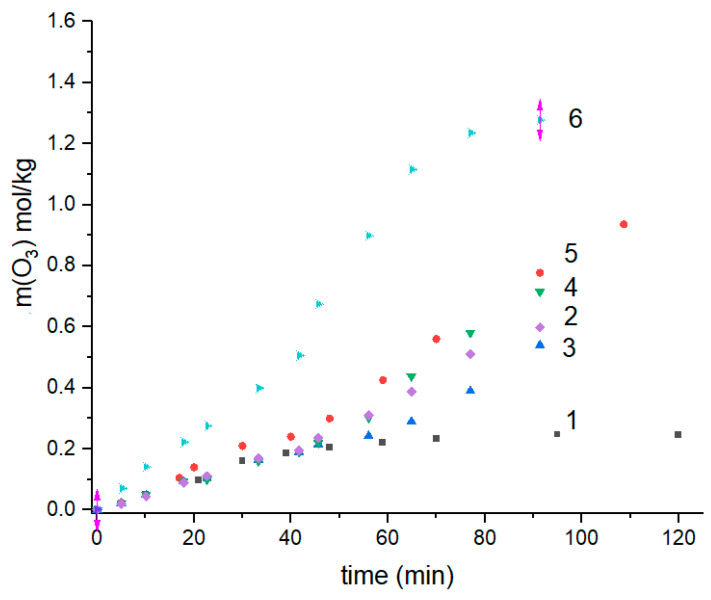
Ozone consumption by the ultrathin fibers at different wt.% of biopolymers PHB:PLA: 1—100:0; 2—0:100; 3—70:30; 4—30:70; 5—10:90; 6—50:50.

**Figure 6 polymers-13-00941-f006:**
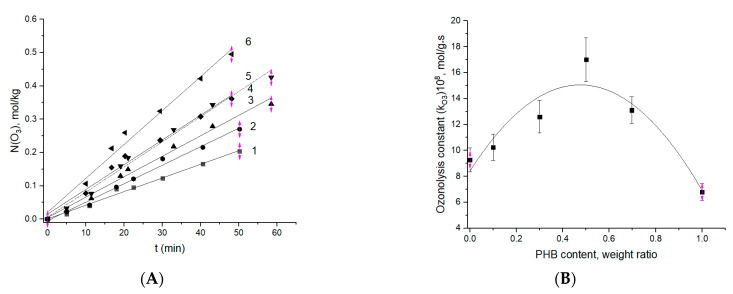
(**A**) Initial stage of ozonolysis for the fibers with different ratios—PHB:PLA (wt.%): 1—100:0; 2—0:100; 3—70:30; 4—30:70; 5—10:90; 6—50:50. (**B**) Effect of PHB content in the fibers on the effective constant ozonolysis.

**Figure 7 polymers-13-00941-f007:**
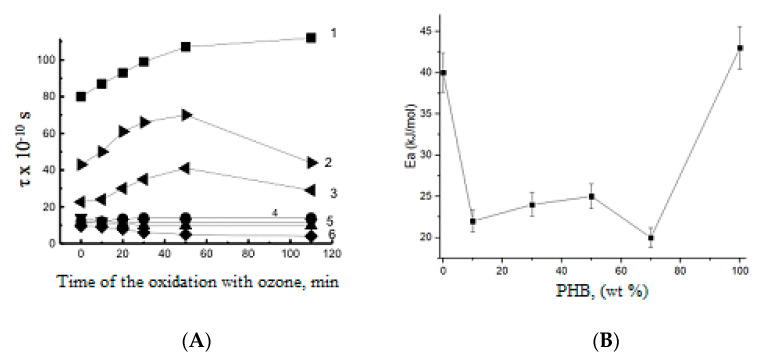
(**A**) The TEMPO correlation time (τ) for the PHB-PLA fibrous mats at different times of ozonation. Polymer ratio—PHB:PLA (wt.%): 1—100:0, 2—0:100, 3—70:30, 4—50:50, 5—30:70, 6—10:90. (**B**) The energy activation of TEMPO probe mobility in the fibrous blends of PHB and PLA.

**Table 1 polymers-13-00941-t001:** The water isotherm constants calculated from linear regression of Equation (4) with corresponding coefficients of determination.

PLA:PHB wt.%	B_1_	B_2_	R^2^
100:0	50.6	0.74	0.988
10:90	27.8	0.66	0.932
30:70	19.2	0.50	0.970
50:50	22.1	0.72	0.974
70:30	15.8	0.36	0.913
0:100	1.25	0.41	0.983

R^2^ is the square of the correlation coefficient (R-squared) for the linear regression.

## Data Availability

The data presented in this study are available on request from the corresponding author.

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
