# Peer review of "Aggressive Impacts Affecting the Biodegradable Ultrathin Fibers Based on Poly(3-Hydroxybutyrate), Polylactide and Their Blends: Water Sorption, Hydrolysis and Ozonolysis"

_polymers, 2021, doi:10.3390/polym13060941_

Round 1
Reviewer 1 Report
Fiber has been widely used in many fields of life. The eco-friendly, fully biodegradable materials, specifically the films with the micrometric diapason of thickness and the ultrathin fibers with the submicrometric diameter, are on the top of the demand for a constructive and special performance. For example, textiles, based on synthetic fibers are supposed to be one of the biggest sources for microplastic. In addition, increasing interest in bio-based polymers and fibers has led to the development of several alternatives to conventional plastics and fibers. Besides, biopolymer fiber can be made from renewable, environmentally friendly resources and be fully biodegradable. Biogenic resources with high content of carbohydrates like starch-containing plants have huge potentials to substitute conventional synthetic plastics in a number of applications. In this paper, Ultrathin electrospun fibers of pristine biopolyesters, poly(3-hydroxybutyrate) and polylactic acid, as well as their blends, were obtained and then explored after exposure to hydrolytic (phosphate buffer) and oxidative (ozone) media. All the fibers were obtained from cosolvent, chloroform, by solution-mode electrospinning. The structure, morphology, and segmental dynamic behavior of the fibers were determined by optical microscopy, SEM, ESR, and others. The topic is important, the results are interesting and the methodology followed is appropriate, while the content falls well within the scope of this Journal. In general the paper makes fair impression and my recommendation is that it merits publication in this Journal, after the following major revision:
- The authors need to reorganize the current introduction, which normally consists of three parts at least: background, literature review, brief of the proposed work. The current one is nothing but a literature review. Why their work is important comparing to previous reports? I think this is essential to keep the interest of the reader.
- In Fig.1 and 6, the authors should give the explanations for the difference of data collected from different sources.
- Materials and Methods part. Although the results look “making sense”, the current form reads like a simple lab report. The authors should dig deeper in the results by presenting some in-depth discussion.
- The fibrous blend PLA/PHB ratio affects the rate of hydrolysis and ozonolysis so that the fibers with prevalent content of PLA display poor resistance to degradation in aqueous and gaseous media. The authors should give some explanation on above results and analyze the physical mechanism in detail.
- Fibrous materials has been widely used in many fields of life. The present work mainly focuses on lab work. It does not necessarily imply that the theoretic work (modeling) is not important. The authors omit this part during the current literature review, which should include a brief review of the theoretic work after revision. In the theoretic perspective, fractal theory is a very important tool, which can be used to investigate the structure, morphology, and segmental dynamic behavior of the fibers (see [A fractal model for capillary flow through a single tortuous capillary with roughened surfaces in fibrous porous media, Fractals, 2021, 29(1):2150017; Fractals, 2019, 27(7): 1950116; Powder Technology, 2019, 349:92-98]). Authors should introduce some related knowledge to readers.
- Please, expand the conclusions in relation to the specific goals and the future work.
Author Response
Dear Reviewer!
Please see the attachment.

Reviewer 2 Report
Manuscript of A.A. Olkhov et al are looking at issues related to the use of poly (3-hydroxybutyrate) and polylactide to produce spinnable biodegradable products. It should be noted that the issues under consideration are relevant, which is associated with active work on the production of new biodegradable materials. The main areas of application of the described materials can be food packaging and medical devices. Based on this, the manuscript will be of interest not only to scientists and specialists in polymer processing, but also to environmentalists, medical expert, etc.
A few minor comments are presented below:
Line 53. "monomers" are best replaced with "polymers".
Line 70. I recommend that the authors replace "article" with "study".
Line 80, 82 onwards. "molding" should be replaced with "spinning".
Line 92. I propose to replace "50 x 30 mm2" with, for example, "0.0015 m2"
Line 189. It is necessary to check the equation number "Eq. 1"
Line 243. Fig. 3. Figure B is missing. Must be added.
Author Response

(The authors gave the same response as above.)

Round 2
Reviewer 1 Report
Ref. 76 should be corrected as “Xiao, B.; Huang, Q.; Chen, H.; Chen, X.; Long, G. A fractal model for capillary flow through a single tortuous capillary with roughened surfaces in fibrous porous media. Fractals 2021, 29, 2150017.”